# Seroreactivity to *Coxiella burnetii* in an Agricultural Population and Prevalence of *Coxiella burnetii* Infection in Ticks of a Non-Endemic Region for Q Fever in South Korea

**DOI:** 10.3390/pathogens10101337

**Published:** 2021-10-17

**Authors:** Jeong-Rae Yoo, Mi-Sun Kim, Sang-Taek Heo, Hyun-Joo Oh, Jung-Hwan Oh, Seo-Young Ko, Jeong-Ho Kang, Sung-Kgun Lee, Woo-Seong Jeong, Gil-Myeong Seong, Hyun-Jung Lee, Chul-Hoo Kang, Ji-Hyun Moon, Keun-Hwa Lee, Sung-Wook Song

**Affiliations:** 1Department of Internal Medicine, College of Medicine, Jeju National University, Jeju 63241, Korea; mdyoojr@jejunu.ac.kr (J.-R.Y.); cadevar@jejunu.ac.kr (S.-T.H.); dfac36@jejunu.ac.kr (W.-S.J.); sgm@jejunu.ac.kr (G.-M.S.); 2Center for Farmers’ Safety and Health, Jeju National University Hospital, Jeju 63241, Korea; junghwan@jejunu.ac.kr (J.-H.O.); c1175@jejunu.ac.kr (S.-Y.K.); siva123456@jejunu.ac.kr (J.-H.K.); casa0140@jejunu.ac.kr (S.-K.L.); sigano@jejunuh.co.kr (H.-J.L.); chulhookang@jejunuh.co.kr (C.-H.K.); jihyunm@jejunu.ac.kr (J.-H.M.); 3Department of Internal Medicine, Jeju National University Hospital, Jeju 63241, Korea; drkms1016@jejunuh.co.kr (M.-S.K.); hjoh27@jejunuh.co.kr (H.-J.O.); 4Department of Neurology, College of Medicine, Jeju National University, Jeju 63241, Korea; 5Department of Emergency Medicine, College of Medicine, Jeju National University, Jeju 63241, Korea; 6Department of Physical Medicine and Rehabilitation, Jeju National University Hospital, Jeju 63241, Korea; 7Department of Neurology, Jeju National University Hospital, Jeju 63241, Korea; 8Department of Family Medicine, College of Medicine, Jeju National University, Jeju 63241, Korea; 9Department of Microbiology, College of Medicine, Hanyang University, Seoul 04763, Korea; yomust7@hanyang.ac.kr

**Keywords:** *Coxiella burnetii*, Q fever, tick-borne disease, seroprevalence, South Korea

## Abstract

*Coxiella burnetii* infects humans and wild and domesticated animals. Although reported cases on Jeju Island, off the coast of South Korea, are rare, the region is considered to have a high potential for Q fever. We investigated the seroprevalence of antibodies to *C. burnetii* in 230 farmers living in ten rural areas on Jeju Island between January 2015 and December 2019. Blood samples were collected and examined for *C. burnetii* Phase I/II IgM and IgG antibodies. Trained researchers collected ticks from rural areas. Clone XCP-1 16S ribosomal RNA gene sequencing was performed to identify *Coxiella* species from the collected ticks. The overall seroprevalence of antibodies to *C. burnetii* in farmers was 35.7%. The seroprevalence was significantly higher in fruit farmers. Of the collected ticks, 5.4% (19/351) of the *Haemaphysalis longicornis* ticks harbored *C. burnetti.* A high seroprevalence of antibodies to *C. burnetii* was observed in this region of Jeju Island, confirming that *C. burnetti* is endemic. Physicians should thus consider Q fever in the differential diagnosis of patients that present with acute fever after participating in outdoor activities.

## 1. Introduction

Query fever (Q fever) is a widespread disease caused by *Coxiella burnetii* [1], an obligate gram-negative intracellular bacterium of the family *Coxiellaecae* [2]. *C*. *burnetii* infects humans and several wild and domesticated animals, and survives in arthropod hosts such as ticks. Domestic ruminants (primarily cattle, sheep, and goats) are the most important reservoir of *C*. *burnetii* [3,4]. *C. burnetii* DNA has been detected in the blood samples of cattle, horses, and goats in South Korea [5].

*C*. *burnetii* is transmitted to humans mainly via the inhalation of contaminated aerosols from infected animals. Other routes of transmission to humans include the consumption of contaminated milk and dairy products, skin or mucosal contact, blood transfusion, and sexual transmission [4,6,7]. The vector capacity of ticks to transmit *C. burnetii* remains unclear [8]. A recent report described a case of severe fever with thrombocytopenia syndrome and Q fever in a girl in South Korea following a tick bite [9], and it has been suggested that *C. burnetii* infection can be transmitted through tick bites [10]. Seroprevalence surveys in human populations have revealed that the seroprevalence is highest (up to 30%) in African countries with a high density of domestic ruminants [4]. Although Q fever has long been considered a rare and regionally localized disease, it has been reported in almost every region worldwide [4]. *C. burnetii* was first isolated from hard ticks (*Dermacentor andersoni* in Montana and *Haemaphysalis humerosa* in Australia), and has been identified in more than 40 hard tick species, 14 soft tick species, and many other arthropods worldwide [4]. Among the tick species, *Haemaphysalis longicornis* is the dominant tick species (88.9%) in South Korea [10]. Although the vector capacity of Korean ticks for transmitting *C. burnetii* is unknown, ticks are known to harbor the organism. In a study of 105 *H. longicornis* ticks conducted in 2004, two were PCR positive for *Coxiella* [11], and in a study of 213 ticks collected from horses on Jeju Island, 121 (52.4%) were PCR positive for *C. burnetii* [12]. In addition, the seroprevalence of *C. burnetii* infection in cattle on Jeju Island is higher than in mainland South Korea [13].

Q fever has been notifiable in South Korea since 2012, and 162 cases were recorded in the national notifiable diseases surveillance system by the Korea Disease Control and Prevention Agency from 2001 to 2019. Jeju Island, which is located off the coast of South Korea, is an area with dense forest and grassland and is home to several wild animals (deer, roe deer, wild boar, badgers, and field mice) and livestock (horses, beef cattle, dairy cattle, and pigs) [14], which creates a favorable environment for ticks. In the past 20 years, only two cases of Q fever have been reported on Jeju Island [15]. Although reported cases of Q fever are rare in this region [15], is the area is considered to have potential for Q fever to occur [10,12] because of the high prevalence of *Coxiella* in ticks on horses [12]. Ticks are considered to be a potential vector for *C. burnetii*, although this has not been confirmed. This study aimed to investigate the seroprevalence of antibodies to *C. burnetii* in an agricultural population and the prevalence of *C. burnetti* infection in ticks on Jeju Island.

## 2. Material and Methods

### 2.1. Study Setting and Participant Selection and Recruitment

This study was conducted from January 2015 to December 2019 in an agricultural population living in the rural areas of Jeju Island. Blood samples were collected from the participants at the Center for Farmers’ Safety and Health at a teaching hospital on Jeju Island with the support of the Safety from Agricultural Injuries in Farmers (SAIF) organization. The island has a humid, subtropical climate (mean summer temperature: 24.7 °C; and mean winter temperature: 7.1 °C) and had a population of 696,000 in 2019, of whom approximately 12% (83,133) were farmers [16].

The study participants were farmers residing on Jeju Island. First, ten rural villages (-myeon and -eup according to the South Korean administrative division system) were selected based on the type of agriculture practiced and the distance from urban areas, and a sample of 500 farmers (approximately 50 farmers per village) was selected. Second, researchers contacted these farmers by telephone, and 423 consented to participate. Third, the researchers visited each village and conducted face-to-face interviews with the participants. Of the 423 farmers who consented to be interviewed, 352 agreed to provide a blood sample and were enrolled in the study. A 10-mL blood sample was collected from each participant in three serum-separating tubes and one ethylenediaminetetraacetic acid tube. Of the samples collected, 230 had an adequate volume for multiple tests. The remaining 122 samples were excluded because of an insufficient serum volume. All participants provided written informed consent.

Data regarding participants’ sociodemographic status, medical history, Charlson comorbidity index (CCI) score, occupation, type of farming, history of tick bites, history of Q fever, and other relevant information were collected by interviewers using a standardized questionnaire. The CCI score was based on participants’ previous disease diagnoses. The collected blood samples were immediately centrifuged at the interview site and the serum was frozen on the same day and stored at −70 °C until further analysis.

### 2.2. Antibody Detection in Sera

The serum samples were sent to Eon Laboratories (Incheon, Korea) to be tested for antibodies to *C. burnetii*, according to the manufacturer’s instructions [17,18]. Each serum sample was tested twice for anti-*C. burnetii* Phase I/II IgM (1:16 dilution) and IgG (1:16 and 1:256 dilutions) using a commercial indirect immunofluorescence antibody (IFA) assay (Focus Diagnostics, Cypress, CA, USA). The relative sensitivity and specificity of this assay have been reported to be 100% [17,18].

### 2.3. Coxiella burnetii Identification in Ticks

Well-equipped trained researchers collected ticks from the natural environment of Jeju Island between June 2016 and January 2019. The tick sampling sites were located in five rural areas: Aewol-eup (AW), Seon Hul-ri (SH), Jeo Ji-ri (JJ), Ha Do-ri (HD), and Bo Mok-ri (BM) (Figure 1). Ticks were manually collected from each site twice a month during the first and third weeks of each month (January to December) by dragging a white cloth through woodlands in each area for 2 h. Tick species and their developmental stages were identified morphologically using an Olympus SD-ILK-200–2 stereomicroscope (Olympus Corporation, Tokyo, Japan, https://www.olympus-lifescience.com, accessed on 10 May 2016) [19]. Total *H. longicornis* nucleic acid was extracted using a QIAamp RNA Mini kit (QIAGEN Inc., Hilden, Germany, https://www.qiagen.com, accessed on 10 May 2016) according to the manufacturer’s instructions. Clone XCP-1 16S ribosomal RNA gene sequencing was used to identify *Coxiella* species in the ticks.

### 2.4. Definitions

A positive Phase I/II IgG or IgM antibody reaction at a dilution of ≥1:16 was considered to be seroreactive [4,20]. The *C. burnetii*-seropositive group (CPG) included participants with positive *C. burnetii* Phase I/II IgM or IgG results, whereas the *C. burnetii*-seronegative group (CNG) included participants with negative results.

### 2.5. Statistical Analysis

Statistical analyses were performed using SPSS 20.0 (IBM Corp., Armonk, NY, USA). The results of categorical variables were summarized as frequencies and proportions. The results of continuous variables were summarized as means with standard deviations or as medians with interquartile ranges. Categorical variables were compared using chi-square tests, and continuous variables were compared using the Mann-Whitney U test. *p*-values of <0.05 were considered to be statistically significant.

## 3. Results

### 3.1. Baseline Characteristics of the Agricultural Population in This Study

A total of 230 farmers participated in this study (1, Gujwa-eup; 49, Aewol-eup; 21, Jocheon-eup; and 12, Hangyeong-myeon in Jeju-si; 56, Namwon-eup; 5, Daechon-dong; 11, Seongsan-eup; 74, Jungmun-dong; and 1, Jungang-dong in Seogwipo-si, Figure 1). Their characteristics are summarized in Table 1.

The mean age was 62.9 years, and 157 (68.3%) were male. Eighty-seven (37.8%) participants had hypertension and 37 (16.1%) had diabetes. The types of farming were fruit (72.6%) and field crops (27.4%). The mean total farming period was 34.2 ± 17.4 years, and the mean time engaged in farming activities per day was 6.2 ± 2.6 h. Eleven farmers were engaged in mixed farming, including crop cultivation and animal husbandry. The types of personal protective equipment used while working included glasses (3.9%), face masks (64.5%), gloves (37.7%), aprons (4.8%), and boots (70.6%).

### 3.2. Seroprevalence of Antibodies to Coxiella burnetii

The overall seroprevalence of IgM or IgG (Phase I and II antibodies) antibodies to *C. burnetii* was 35.7% (Table 2). The seroprevalence of Phase I and Phase II IgG antibody titers to *C. burnetii* ≥1:16 was 15.7% and 23.0%, respectively; the seroprevalence of Phase I and Phase II IgG antibody titers to *C. burnetii* ≥1:256 was 1.3% and 3.9%, respectively; and the seroprevalence of Phase I and Phase II IgM antibody titers to *C. burnetii* ≥1:16 was 13.9% and 17.0%, respectively.

The areas with the highest seroprevalence of antibodies to *C. burnetii* were Jocheon-eup in Jeju-si (11/22, 50%) and Jungmun-dong in Seogwipo-si (37/74, 50%), followed by Daecheon-eup (2/5, 40%), Namwon-eup (16/56, 29%), Hangyeong-myeon (3/12, 25%), Aeawol-eup (12/49, 25%), and Seongsan-eup (1/11, 9%). Neither participant in Gujwa-eup in Jeju-si (0/1) or Jungang-dong in Seogwipo-si (0/1) was seropositive for antibodies to *C. burnetii*.

### 3.3. Comparison of the Clinical Characteristics of the Coxiella burnetii-Positive and -Negative Groups

Of the 230 participants, 82 were included in the CPG and 148 were included in the CNG (Table 1). The mean participant age was 59.5 ± 14.7 years and 55.3 ± 19.8 years in the CPG and CNG, respectively, and 56.5% and 50.0% of the CPG and CNG, respectively, were male. The seroprevalence of antibodies to *C. burnetii* was higher in older participants than in younger participants, but the difference was not statistically significant (*p* = 0.07). There was no significant association between anti-*C. burnetii* antibody status and comorbidities or history of tick bites. Fruit-farming was significantly more common in the CPG than in the CNG (81.7% vs. 67.6%, *p* = 0.02). The total farming period, daily farming period, and possession of livestock did not differ significantly between the two groups. No participants in either group reported consuming raw milk. In addition, the types of personal protective equipment used did not differ significantly between the two groups.

### 3.4. Coxiella burnetii in Ticks on Jeju Island

In total, 3193 ticks were collected. Of the total ticks collected, *H. longicornis* was the most common (98.5%), followed by *Haemaphysalis flava* (1.5%) (Appendix A). *C. burnetii* was detected in 19/354 (5.4%) of the *H. longicornis* ticks tested (Table 3 and Appendix A). The *C. burnetii*-infected ticks were from Seon Hul-ri (5), Ha Do-ri (4), Jeo Ji-ri (4), Aewol-eup (3), and Bo Mok-ri (3). Of the 19 *C. burnetii*-infected ticks, the majority (15/19) were collected in the winter months (12 in January, two in March, two in October, two in November, and one in December).

## 4. Discussion

In this study, the seroprevalence of antibodies to *C. burnetii* was 37.5% in farmers. The seroprevalence of antibodies to *C. burnetii* was highest among fruit farmers living in the southwestern part of Jeju Island, and was higher in older participants than in younger participants.

Only two cases of Q fever on Jeju Island were reported in the Korean national notifiable infectious diseases system during the period 2001–2020 [15]; however, based on the high seroprevalence of antibodies to *C. burnetii* found in this study, *C. burnetii* infection is endemic in the region. A diagnosis of *C. burnetii* can only be confirmed by performing an IFA test (Phase I and Phase II) in the Department of Laboratory at the Korea Disease Control and Prevention Agency (KDCPA). Physicians send a sample to the KDCPA with a patient declaration form, and they receive the result a few weeks later. Physicians may find it difficult to diagnose Q fever in patients with acute febrile illnesses because there are no readily available commercial diagnostic tests for *C. burnetii* infection, and there is a lack of seroepidemiological data in South Korea, so the level of awareness among physicians is low.

The seroprevalence in these farmers was considerably higher than that reported in previous seroepidemiological surveys in another asymptomatic rural population (1.5%) and in cattle slaughterhouse workers (9.1%) in South Korea [21,22]. The high seroprevalence was unexpected because the agricultural workers have less opportunity to being exposed to *C. burnetii*-containing aerosols compared with slaughterhouse workers who may be exposed to feces, urine, milk, and vaginal discharge of animals infected with *C. burnetii* during the slaughtering process [6]. *C. burnetii*-infected ticks may remain infected for an extended time period–between 200 and 1000 days and more than 10 years in several cases–and the excretion of *C. burnetii* in tick feces occurs on the skin of the animal host while feeding. *C. burnetii* may persist for prolonged periods in the soil, and bacterial aerosols can be delivered for at least 30 km through air [4]. For the above reasons, most routes of human infection occur after the inhalation of aerosols containing *C. burnetii* from infected animals and their products. Individuals with Q fever can develop the disease without animal contact because transmission to humans is influenced by geographical factors such as the subtropical climate, the specific natural environment, and the heavy wind on Jeju Island [23].

Physicians may not consider Q fever in the differential diagnosis in patients with acute febrile illness because the symptoms and signs of Q fever are nonspecific; acute fever is common due to severe fever with thrombocytopenia syndrome (in the spring and the summer) and scrub typhus (in the autumn) in individuals who participate in outdoor activity [23,24,25,26], and most infected individuals have no history of animal contact or occupational exposure [27].

In recent published reports, of the individuals diagnosed with Q fever, 2.0% (48/2434) were diagnosed with acute Q fever endocarditis, and *C. burnetii*-associated interstitial lung disease was rare [28,29]. In addition, *C. burnetii* infection may be associated with lymphadenitis and cholecystitis [30]. On Jeju Island, transthoracic echocardiography should be performed in patients with confirmed *C. burnetii* infection and in patients with unexplained acute fever [31,32], because Q fever can lead to severe complications. Fruit farmers and inhabitants of Jocheon-eup and Jungmun-dong were at particularly high risk of infection in this study. Patients with acute febrile illness who reside in these areas should have a diagnostic assessment for Q fever.

The composition of the agricultural population of Jeju Island differs from that on the mainland of South Korea. Among farmers of Jeju Island, the proportion of fruit farmers is relatively high and that of livestock farmers is relatively low (62.5% and 2.2%, respectively) compared with the mainland farming population (16.9% and 5.3%, respectively) [16].

Another recent study conducted on Jeju Island found a high seroprevalence (52.4%) of antibodies to *Coxiella* in horses [12]. Jeju Island is known for its high number of horses. Its geographical location and climate make it ideal for horse breeding, and 56.0% of the horses bred nationally in South Korea are bred on Jeju Island [14]. There are approximately 390,000 horseback riders living on Jeju Island. Horseback riders may have a high risk of *C. burnetii* infection from contact with infected horses or aerosols of horse feces.

Slaughterhouse workers and veterinarians are generally at high risk of *C. burnetii* infection, and most individuals diagnosed with Q fever live in rural areas [10,22,33,34]. Q fever is most commonly diagnosed in the summer and the early autumn [27]. In this study, participants in the CPG were may have been infected by a tick bite or by droplet transmission from an infected animal because the agricultural population did not wear N95 masks or other personal protective equipment during the pre-coronavirus disease pandemic era, and infected domestic animals shed a large amount of *C. burnetii*. In addition, farmers generally do not wear face masks after their work is finished, and they may have become infected via droplet infection in their natural environment. Protective masks (N95 or higher type) can provide better protection against *C. burnetii* infection compared to standard surgical masks and personal protective equipment [35].

In the CPG, 17% of the participants were positive for Phase II IgM antibodies, so they may have had an acute *C. burnetii* infection at the time of the survey. Participants did not have a fever within the two weeks before sample collection, but *C. burnetii* infection in humans is usually asymptomatic or presents as a mild disease with spontaneous recovery. However, we were unable to diagnose acute *C. burnetii* infection in study participants because we did not collect serial samples and could not compare the IgM titer between Phases I and II. The median time from symptom onset to serological diagnosis is 21 days, and 60% of cases in South Korea are diagnosed between June and September [27].

Studies conducted in other countries have shown a seroprevalence of 15–30% in African countries [34,35,36,37,38,39,40]; approximately 30.0% in Iran [41]; 5.3% in the rural population of Australia [42]; 2.4% in the Netherlands [43]; 24% among patients with acute febrile illness in France [4], 1.3% in Kazakhstan [44], and 5.8% in Asia [45]. In Japan, the incidence rate of Q fever increased significantly during the period 2000–2004, with a seroprevalence of antibodies to *C. burnetii* of 22–36% in veterinarians [46]. The study found that Q fever had been underdiagnosed for many years in Japan. In previous seroprevalence surveys conducted in South Korea, the seroprevalence of antibodies to *C. burnetii* was 9.1% in cattle slaughterhouse workers and 11.0% in dairy cattle farmers [22,47], and the seroprevalence of Phase II IgM and IgG antibody titers to *C. burnetii* infection of ≥1:16 was 9.1% and 0.2%, respectively. Although there have not been any recent reports of cases of Q fever on Jeju Island, this study provides novel evidence that the seroprevalence of antibodies to *C. burnetii* is relatively high in the agricultural population of Jeju Island, and there is thus a potential for Q fever.

This study has some limitations. First, we did not perform polymerase chain reaction analysis to determine whether participants in the CPG had a *C. burnetii* infection, as the participants did not have symptoms of Q fever and the volume of sample collected was insufficient for performing further tests. Second, we did not evaluate the seroprevalence of antibodies to *C. burnetii* in inhabitants of urban areas of Jeju Island. Third, we did not test the samples for cross-reactivity with other infectious pathogens. However, according to the manufacturer’s instructions, neither Phase I nor Phase II *C. burnetii* antigen cross-reacts with other *Rickettsia* or bacteria, so false-positive reactions are unlikely [17,18]. Fourth, we did not investigate participants’ history of contact with various animals or the type of compost and manure to which they were exposed. Finally, this was not a diagnostic study of primary *C. burnetii* infection and persistent *C. burnetii* infection. Because Phase II antibodies are detectable within 7–15 days after the onset of clinical symptoms and then decrease within three to six6 months, antibodies are detectable by the third week after infection in 90% of infected individuals [35,48]. The prevalence of anti-*C. burnetii* antibody titers of Phase I/II IgM ≥ 1:16, Phase I/II IgG ≥ 1:16, and Phase I/II IgG ≥ 1:256 were analyzed in this cross-sectional study, but we did not perform serial testing to assess changes in the antibody titers.

In conclusion, Jeju Island has been considered to be a non-endemic area for Q fever, and little is known about the incidence of *C. burnetii* infection in South Korea or of the viability of ticks as possible vectors. Previous studies have not focused on the agricultural population of South Korea, who have many opportunities for tick bite exposure. The study revealed that the agricultural population of Jeju Island has a higher seroprevalence of antibodies to *C. burnetii* than in regions where Q fever is considered to be endemic, and that *C. burnetii* is prevalent in ticks on Jeju Island. Therefore, physicians should consider Q fever in the differential diagnosis of patients with acute febrile illnesses and a history of outdoor activity. Further studies of *C. burnetii* seroprevalence are necessary in South Korea, and active surveillance and monitoring are needed on Jeju Island.

## Figures and Tables

**Figure 1 pathogens-10-01337-f001:**
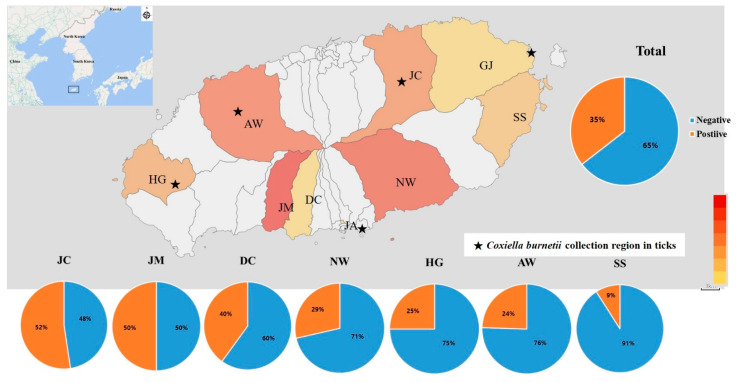
Geographic distribution of seroreactivity in participants with *Coxiella burnetii* in an agricultural population and *Coxiella burnetii*-positive ticks in Jeju Island. The inset shows the location of Jeju Island near of the coast of South Korea. The orange to red shading of the regions indicates the number of participants in this study from each region. In the pie charts, orange indicates the seroprevalence of antibodies to *Coxiella burnetii*. The asterisks indicate the tick collection sites. The tick collection sites were located in five rural areas: Aewol-eup (AW), N 33° 45′ 90.0″, E 126° 34′ 24.4″; Seon Hul-ri (SH), N 33° 50′ 82.6″, E 126° 69′ 92.9″; Jeo Ji-ri (JJ), N 33° 34′ 00.2″, E 126° 26′ 45.8″; Ha Do-ri (HD), N 33° 50′ 69.6″, E 126° 89′ 05.3″ and Bo Mok-ri (BM), N 33° 24′ 83.1″, E 126° 60′ 32.2″. Abbreviations: AW, Aewol-eup; BM, Bo Mok-ri; DC, Daechoen-dong HG, Hangyeong-meon; JM, Jungmun-dong; JC, Jocheon-eup; JJ, Jeo Ji-ri; NW, Nanwon-eup; SS, Seongsan-eup; GJ. Gujwa-eup; JA, Jungang-dong.

**Table 1 pathogens-10-01337-t001:** Baseline characteristics of agricultural population in this study (*n* = 230).

Variables	All (*n* = 230)	CNG (*n* = 148)	CPG (*n* = 82)	*p*
Sex, male (*n*, %)	157 (68.3)	103 (69.6)	54 (65.8)	0.559
Age (mean ± SD)	62.9 ± 11.9	62.4 ± 12.6	63.7 ± 10.7	0.424
<40 years	9 (3.9)	7 (4.7)	2 (2.4)	0.074
40 to 49 years	28 (12.2)	23 (15.5)	5 (6.1)	
50 to 59 years	47 (20.4)	23 (15.5)	24 (29.3)	
60 to 69 years	68 (29.6)	42 (28.4)	26 (31.7)	
70 to 79 years	66 (28.7)	45 (30.4)	21 (25.6)	
≥80 years	12 (5.2)	8 (5.4)	4 (4.9)	
Comorbidities (*n*, %)				
Hypertension	87 (37.8)	56 (37.8)	31 (37.8)	0.996
Diabetes	37 (16.1)	25 (16.9)	12 (14.6)	0.711
Kidney disease	1 (0.4)	0 (0)	1 (1.2)	0.357
Liver disease	1 (0.4)	0 (0)	1 (1.2)	0.357
Stroke	3 (1.3)	2 (1.4)	1 (1.2)	1.000
Married (*n*, %)	228 (99.1)	147 (99.3)	81 (98.8)	0.587
Household member (*n*, %)				0.492
1 generation	137 (59.6)	91 (61.5)	46 (56.1)	
2 generations	62 (27.0)	36 (24.3)	26 (31.7)	
3 generations	31 (13.5)	21 (14.2)	10 (12.2)	
Type of farming (*n*, %)				0.022
Fruit farming	167 (72.6)	100 (67.6)	67 (81.7)	
Field	63 (27.4)	48 (32.4)	15 (18.3)	
Total farming period, years	34.2 ± 17.4	34.2 ± 17.4	34.2 ± 17.4	0.47
Farming period per day, hours	6.2 ± 2.6	6.1 ± 2.6	6.2 ± 2.5	0.73
withs farming equipment (*n*, %)	204 (88.7)	131 (88.5)	73 (89.0)	1.00
with greenhouse (*n*, %)	122 (53.0)	122 (53.0)	39 (47.6)	0.22
with an animal shed (*n*, %)	11 (4.8)	8 (5.4)	8 (5.4)	0.55
Wearing protective equipment (*n*, %)				
Glasses	9 (3.9)	7 (4.7)	2 (2.4)	0.221
Mask	149 (64.5)	94 (63.1)	55 (67.1)	0.291
Gloves	87 (37.7)	57 (38.3)	30 (36.6)	0.261
Apron	11 (4.8)	8 (5.4)	3 (3.7)	0.255
Boots	163 (70.6)	99 (66.4)	64 (78.1)	0.105

Values are presented as mean ± standard deviations. Abbreviations: n, number; SD, standard deviation; CPG, *Coxiella burnetiid*-seropositive group; CNG, *Coxiella burnetii*-seronegative group.

**Table 2 pathogens-10-01337-t002:** Seroreactivity of *Coxiella burnetii* in an agricultural population on Jeju Island.

Immunogloblin	IgM		IgG			
IFA titer	1:16		1:16		1:256	
Phase	Phase I	Phase II	Phase I	Phase II	Phase I	Phase II
Results	13.9% (32/230)	16.9% (39/230)	15.6% (36/230)	23.0% (53/230)	1.3% (3/230)	3.9% (9/230)
Sum *	20.0% (46/230)		23.0% (53/230)		3.9% (9/230)	
Total sum *	35.6% (82/230)					

Abbreviations: Ig, immunoglobulin; IFA, indirect immunofluorescence antibody assay. * a positive Phase I/II IgG or IgM antibody reaction at a dilution of ≥1:16 was considered to be seroreactive.

**Table 3 pathogens-10-01337-t003:** Identification of *Coxiella burnetii* in ticks on Jeju Island during 2016–2018.

Sample No.	Location	YY-MM-DD	Species of Tick	Developmental Stage of Tick	BLAST
4th-A6	SH	2016-08-26	*H.long*	Adult (F)	Uncultured *Coxiella* sp. clone XCP-1 16S ribosomal RNA gene, partial sequence (99%)
4th-A7	SH	2016-08-26	*H.long*	Adult (F)	Uncultured *Coxiella* sp. clone XCP-1 16S ribosomal RNA gene, partial sequence (92%)
10th-G6	BM	2016-12-08	*H.long*	Nymph	Uncultured *Coxiella* sp. clone XCP-1 16S ribosomal RNA gene, partial sequence (89%)
12th-E11	HD	2017-01-10	*H.long*	Nymph	Uncultured *Coxiella* sp. clone XCP-1 16S ribosomal RNA gene, partial sequence (99%)
16th-A4	SH	2017-03-09	*H.long*	Nymph	Uncultured *Coxiella* sp. clone XCP-1 16S ribosomal RNA gene, partial sequence (99%)
16th-A5	SH	2017-03-09	*H.long*	Nymph	Uncultured *Coxiella* sp. clone XCP-1 16S ribosomal RNA gene, partial sequence (99%)
30th-C4	JJ	2017-11-01	*H.long*	Nymph	Uncultured *Coxiella* sp. clone XCP-1 16S ribosomal RNA gene, partial sequence (99%)
35th-C2	JJ	2018-01-18	*H.long*	Nymph	Uncultured *Coxiella* sp. clone XCP-1 16S ribosomal RNA gene, partial sequence (99%)
35th-C4	JJ	2018-01-18	*H.long*	Nymph	Uncultured *Coxiella* sp. clone XCP-1 16S ribosomal RNA gene, partial sequence (99%)
35th-C6	JJ	2018-01-18	*H.long*	Nymph	Uncultured *Coxiella* sp. clone XCP-1 16S ribosomal RNA gene, partial sequence (99%)
35th-E4	HD	2018-01-18	*H.long*	Nymph	Uncultured *Coxiella* sp. clone XCP-1 16S ribosomal RNA gene, partial sequence (99%)
35th-E9	HD	2018-01-18	*H.long*	Nymph	Uncultured *Coxiella* sp. clone XCP-1 16S ribosomal RNA gene, partial sequence (99%)
35th-E10	HD	2018-01-18	*H.long*	Nymph	Uncultured *Coxiella* sp. clone XCP-1 16S ribosomal RNA gene, partial sequence (99%)
35th-G3	BM	2018-01-18	*H.long*	Nymph	Uncultured *Coxiella* sp. clone XCP-1 16S ribosomal RNA gene, partial sequence (99%)
35th-G4	BM	2018-01-18	*H.long*	Nymph	Uncultured *Coxiella* sp. clone XCP-1 16S ribosomal RNA gene, partial sequence (99%)
35th-A1	AW	2018-01-18	*H.long*	Nymph	Uncultured *Coxiella* sp. clone XCP-1 16S ribosomal RNA gene, partial sequence (99%)
35th-A2	AW	2018-01-18	*H.long*	Nymph	Uncultured *Coxiella* sp. clone XCP-1 16S ribosomal RNA gene, partial sequence (99%)
35th-A3	AW	2018-01-18	*H.long*	Nymph	Uncultured *Coxiella* sp. clone XCP-1 16S ribosomal RNA gene, partial sequence (99%)
37th-A4	SH	2017-11-06	*H.long*	Nymph	Uncultured *Coxiella* sp. clone XCP-1 16S ribosomal RNA gene, partial sequence (99%)

Abbreviations: No, number; YY-MM-DD; year-month-day; *H.long*, *Haemaphysalis longicornis*; M, male; F, female; BLAST, Basic Local Alignment Search Tool; sp., species; RNA, ribonucleic acid; AW, Aewol-eup; SH, Seon Hul-ri; JJ, Jeo Ji-ri; HD, Ha Do-ri; BM; Bo Mok-ri.

## Data Availability

The data are not publicly available due to privacy of the patients.

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
