# Peer review of "Seroreactivity to Coxiella burnetii in an Agricultural Population and Prevalence of Coxiella burnetii Infection in Ticks of a Non-Endemic Region for Q Fever in South Korea"

_pathogens, 2021, doi:10.3390/pathogens10101337_

Round 1
Reviewer 1 Report
This is an interesting and well written article.
Just some remarks
-There is no correlation between Q fever positive serology and clinical history of hepatitis pneumonia and this is lacking. Maybe could you complete with a chapter on clinical correlation among seropositive people and with a table.
-A table with the titers of IgG and IgM phase I and phase II would be helpful to determine the serological profile of positive individuals. Please cite Fournier, 1998
-If high levels of phase I IgG are detected, what public health strategy could you propose to detect persistent, pauci-symptomatic C. burnetii infection? What is the interest of a seroprevalence study?
-In your conclusion, you should emphasize the severity of C. burnetii infection, especially when the cardiovascular system is affected (please quote Melenotte, JAMA Network, 2018; Melenotte Clin Infect Dis 202 acute Q fever endocarditis) and the pulmonary sequelae that have been reported (Melenotte, Clin infect Dis C. burnetii the hidden pathogen of interstitial lung disease"). Are these disease endemic in this region?
-Finally, you have focused on ticks, but the main route of contamination in humans is through aerosols. Do you have any idea of the prevalence of infection in sheep and goats in this region of China?
-do you have a project to study the placenta of the aborted animals
At last, do you have an idea of the C. burnetii strain present in this part of the world?
Author Response
Author response
Manuscript number: Manuscript ID: pathogens-1337633
Manuscript Title: Seroprevalence of Coxiella burnetii antibodies in a healthy agricultural population and prevalence of Coxiella burnetii infection in ticks of non-endemic regionregion for Q fever South Korea
Dear Main Editor and Reviewers:
Thank you for reviewing our manuscript and for your comments.
Response to your comments
Reviewer 1
Comments and Suggestions for Authors
This is an interesting and well written article.
Just some remarks
-There is no correlation between Q fever positive serology and clinical history of hepatitis pneumonia and this is lacking. Maybe could you complete with a chapter on clinical correlation among seropositive people and with a table.
Response: Thank you for your comments. This study enrolled healthy farmers. We asked a question about infectious disease within the past three years in a face-to-face interview. The participants did not have a history of infectious diseases. In addition, even if the participants had had a history of hepatitis or pneumonia, we would not have been able to determine the etiology because we did not have access to their medical records.
-A table with the titers of IgG and IgM phase I and phase II would be helpful to determine the serological profile of positive individuals. Please cite Fournier, 1998
Response: Thank you for your comments. These titers were already provided in Figure 2 and Section 3.2 of the results. However, we changed the figure to table form based on your comments (Table 2, Line 235). We added the reference by Fournier as Reference 41.
-If high levels of phase I IgG are detected, what public health strategy could you propose to detect persistent, pauci-symptomatic C. burnetii infection? What is the interest of a seroprevalence study?
Response: Thank you for your questions. According to modified Duke criteria, a diagnosis of Q fever endocarditis requires a Phase I IgG titer of ≥1:800. The seroprevalence of Phase I IgG ≥1:256 was 1.3%, and we did not test the samples for Phase I IgG titers of ≥1:800.
First, we have a strategy of conducting Q fever diagnostic workups in patients with acute unexplained fever. The persistence of high levels of anti-phase I antibodies despite appropriate treatment, or the reappearance of such antibodies should raise suspicion of possible chronic Q fever. We added some wording to the conclusion section about the need for active assessment for Q fever (Lines 216-217). In cases of confirmed acute Q fever, an immunofluorescence assay should be performed monthly for at least 6 months. The follow-up of patients treated for chronic Q fever should also include serological monitoring.
-In your conclusion, you should emphasize the severity of C. burnetii infection, especially when the cardiovascular system is affected (please quote Melenotte, JAMA Network, 2018; Melenotte Clin Infect Dis 202 acute Q fever endocarditis) and the pulmonary sequelae that have been reported (Melenotte, Clin infect Dis C. burnetii the hidden pathogen of interstitial lung disease"). Are these diseases endemic in this region?
Response: Thank you for your comments and for the references. We have described Q fever endocarditis and pulmonary sequelae in the Discussion section (Lines 165-169). Q fever cases have been reported since 2012 in South Korea, and 162 Korean patients with Q fever have been reported in the national notifiable diseases surveillance system by the Korea Disease Control and Prevention Agency during 2001-2019, but cases of Q fever endocarditis are very rare in our region. In recent literature in South Korea, Jan et al (2018) reported that of 40 patients with blood culture negative infective endocarditis who underwent cardiac valve surgery 8 had positive using C. burnetii PCR assay in Asan Medical Center in South Korea from 2001 to 2016. Cardiovascular infections were the main fatal complications, highlighting the importance of routine screening for valvular heart disease and vascular anomalies in patients with acute Q fever. We have included the references by Melenotte et al. as References 23, 24, and 25,
-Finally, you have focused on ticks, but the main route of contamination in humans is through aerosols. Do you have any idea of the prevalence of infection in sheep and goats in this region of China?
Response: Thank you for your comments. We know that the main route of infection in humans is through aerosol. The study participants were from an agricultural population with a relatively low contact frequency with livestock. We assessed other transmission routes because the local population had many opportunities for exposure to ticks. Particularly, the study region has the highest incidence of severe fever with thrombocytopenia syndrome in South Korea. In the study area the number of horses and pigs is very high, but there are few sheep or goats. However, recent studies have shown that the overall seroprevalence was 10.5% in cattle and 19.1% in Korean native goats. In previous studies, the C. burnetii sequences found in beef and dairy cattle were clustered with several strains of C. burnetii obtained from ticks, humans, goats, and cattle from other countries. (Hwang et al. 2020, Pathogens 9(11):890; not cited). On Jeju Island, 52.4% of horses are seropositive for Coxiella (Reference 10; Lines 175-176), The seroprevalence of C. burnetii in cattle on Jeju Island (21.3%,) is higher than in any of the other three geographical regions of South Korea (reference 11). On Jeju Island in 2019, the number of cases of infection reported in animals was 694 in beef cattle, 45 in dairy cattle, 877 in cattle (type not specified), 278 in pigs, 3 in sheep, and 22 in deer.
-do you have a project to study the placenta of the aborted animals?
Response: Thank you for your question. There are currently no plans, but we will discuss this with staff in the Department of Veterinary Sciences.
At last, do you have an idea of the C. burnetii strain present in this part of the world?
Response: We do not know the most prevalent local strain of C. burnetii. We think that the prevalent strain of C. burnetii is probably zoonotic and pathogenic. Q fever is a very underdiagnosed disease. Further molecular epidemiological studies are needed to investigate the genetic diversity of this bacterium in humans and animals.
Reviewer 2 Report
Paper entitled Seroprevalence study of Coxiella burnetii in a healthy agricultural population of a rare region of Q fever in South Korea describes prevalence of seroreactivity against C. burnetii antigens in residents of Jeju island and prevalence of C. burneti infection in ticks collected from the same territory.
Although the subject is of public health importance, there are several point about which I am concerned.
The title:
- The title of the paper is misleading and not giving any hints that tick analysis was conducted
- I don’t understand what the authors wanted to say with the term rare region of Q fever. Is that region endemic or non-endemic?
Introduction:
Line 46-47: Survival of some microorganism within host does not make the same host a vector for the disease. Please rephrase the sentence.
Line 54: Authors states that C. burnetti are transmitted via tick bite with citing references 6 and 7. Authors in reference 6 stated Humans rarely, if ever, acquire disease through tick bites. In reference 7, there are no information about tick-related transmission of C. burnetti. I ask for authors to give supportive background for their statement.
Line 66-67: has several wild animals and livestock animals, which is a favorable environment for ticks.
Please be more specific. This kind of sentence does not give any relevant information.
Line 69-70: Authors state: Although this region had no reported cases of Q fever, it is considered as a region with a high potential of Q fever, while citing reference 11.
While reading reference 11 I could not find any information related to “high potential of Q fever’’ in Jeju Island. I ask for authors to give supportive background for their statement.
The aim:
The aim is not in alignment with the results described in paper. Aim is oriented only toward serosurvey, while tick sampling and analysis was conducted also.
M & M
In the first step, ten rural villages were selected depending on the type of agriculture practiced and the distance from urban areas.
Are there any urban villages in South Korea? Please be more specific about criteria that you referred as “agriculture practiced” and “distance from urban areas”.
Authors stated that from 352 farmers who accepted to participate in the study 270 samples was acquired. What was the reason for other subjects to be excluded from the study? How much blood was collected and in what kind of vials?
Data regarding patients’ sociodemographic status, medical history, occupation, type of farming, history of tick bites, history of Q fever, history of contact with Q fever, and other relevant information were collected using standardized questionnaires.
I ask for authors to upload used questionnaire as supplementary file.
The collected blood samples were preserved at -70°C after centrifugation, until further analyses.
I propose that authors referrer to centrifuged blood sample as serum or plasma, depending on vials used. How much time has been passed from blood collection to sera/plasma separation and deep freezing?
Line 103-104
The sensitivity and specificity of this laboratory test (Phase I and II) were 100% and 103 100%, respectively [13, 14].
The authors are giving misleading statements. In the assay datasheet manufacturer is referring to Relative Sensitivity and Relative Specificity. In addition, both for IgM and IgG in the manufacturer datasheet it is stated: seroconversion alone may not be diagnostic of current or recent infection.
Coxiella burnetii identification in ticks
- On what locations and in which part of the year the ticks were collected?
- Using witch “manual“ method?
- What key was used for morphological determination?
- What nucleic acid extraction method was used?
- If 3193 ticks were collected, what was the rationality for further examination of 351 tick?
In total, 3,193 ticks were collected, 351 of whom were identified to have severe fever with thrombocytopenia syndrome viruses (SFTSV).
This sentence does not belong to MM section. In addition, I don’t know what is the relevance of SFTSV for this study?
Results
Figure 1 is named Geographic distribution of seroactivity participants with Coxiella burnetii in agricultural population and Coxiella burnetii-positive ticks in Jeju Island.
- Seroreactiviy instead seroactivity
- Authors marked with a star ‘’C. burnetti collection region in ticks’’. I can’t understand what authors wanted to say with this. Please be more specific.
- As I can notice in the region map, all courtiers bordering South Korea except Democratic People's Republic of Korea are labeled. Please label all bordering counties.
Comparison of clinical characteristics between the C. burnetii-positive and -negative groups
Line 161,162 – change term patient to participant
Older farmers showed a trend of a higher risk of C. burnetii infection than 163 younger farmers (p < 0.07)
If authors are referring to risks, I have to ask for them firstly to conduct a risk analysis.
I would ask for authors to explain why are they are assuming that older farmers have a higher risk of C. burnetii infection that younger ones when there is no statistically significant difference between these two age groups?
The mean Charlson comorbidity index (CCI) score did not show any significant difference… Utility of this index was not mentioned in MM section.
Coxiella burnetii in ticks in Jeju Island
Please provide supplementary table with data related to location, time, species of tick, developmental stage of tick and molecular findings for all collected/analyzed ticks.
Discussion
- Physicians may find it difficult to diagnose Q fever in patients with acute febrile illness because there are no easy commercial diagnostic tools and there is a lack of seroepidemiological data in South Korea
I would ask for authors to explain this statement. As for my current knowledge - backbone of effective diagnostic approach are not easy commercial diagnostic tools, neither seroepidemiological data. Determination of endemic areas, surveillance and implementation of specific management protocols are more effective in most of the counties.
-Authors are building discussion according to opinion that all seroreactive patients had C. burnetti infection.
As stated by manufacturer, if patient did not have specific clinical manifestations, C. burnetti infection that cannot be assumed via mentioned tests. I strongly suggest for authors to critically review their results and rewrite discussion with emphasis of explaining the current situation concerning possible exposure to C. burnetti, especially in this case where authors report to have so high seroprevalence compared to other regions/countries, thus without many actual reported Q-fever cases.
Author stated that:
In this study CPG was presumed to have infection due to a tick bite or droplet transmission from an infected animal because the agricultural population did not wear an N95 mask or any other personal protective equipment during the pre-coronavirus disease pandemic era, and infected domestic animals shed a large amount of C. burnetii.
Thus in results they reported that more that 60% of seropositive participants used facial mask. Please explain this discrepancy.
According to current manuscript, this study has many limitations, while the biggest one is non-critical interpretation of serological assays.
Author Response
Author response
Manuscript number: Manuscript ID: pathogens-1337633
Manuscript Title: Seroprevalence of Coxiella burnetii antibodies in a healthy agricultural population and prevalence of Coxiella burnetii infection in ticks of non-endemic region for Q fever South Korea
Dear Main Editor and Reviewers:
Thank you for reviewing our manuscript and for your comments.
Response to your comments
Reviewer 2
Comments and Suggestions for Authors
Paper entitled Seroprevalence study of Coxiella burnetii in a healthy agricultural population of a rare region of Q fever in South Korea describes prevalence of seroreactivity against C. burnetii antigens in residents of Jeju island and prevalence of C. burnetii infection in ticks collected from the same territory.
Although the subject is of public health importance, there are several points about which I am concerned.
The title:
- The title of the paper is misleading and not giving any hints that tick analysis was conducted
- I don’t understand what the authors wanted to say with the term rare region of Q fever. Is that region endemic or non-endemic?
Response: Thank you for your comments. We changed title to “Seroprevalence of Coxiella burnetii antibodies in a healthy agricultural population and prevalence of Coxiella burnetii infection in ticks of a non-endemic region for Q fever in South Korea”
Introduction:
Line 46-47: Survival of some microorganism within host does not make the same host a vector for the disease. Please rephrase the sentence.
Response: Thank you for your comments. We have corrected the sentence. We changed “vector” to “reservoir” (Line 44).
Line 54: Authors states that C. burnetti are transmitted via tick bite with citing references 6 and 7. Authors in reference 6 stated Humans rarely, if ever, acquire disease through tick bites. In reference 7, there are no information about tick-related transmission of C. burnetti. I ask for authors to give supportive background for their statement.
Response: Thank you for your comments. We mentioned various modes of transmission of Q fever in this sentence, and mentioned tick bite as a possibility. This is reported that the detection of C. burnetii coinfection with other arthropod-borne pathogens in ticks. Coinfection with Rickettsia of the spotted fever group and C. burnetii was observed in 5% of Rhipicephalus turanicus ticks. Double infection with C. burnetii and Borrelia burgdorferi was detected in 7 (21%) of Ixodes ricinus ticks, and a positive statistically significant correlation between these two pathogens was observed. These findings of different microorganisms in ticks suggest a common mode of transmission for both human pathogens via arthropods. We added a reference 4, and changed “tick bite” to “ticks” (Lines 46-48).
Line 66-67: has several wild animals and livestock animals, which is a favorable environment for ticks. Please be more specific. This kind of sentence does not give any relevant information.
Response: Thank you for your comments. Of livestock animals, horses, beef cattle, dairy cattle, and pigs are the most common reservoirs in the region. Of wild animals, deer, roe deer, wild boars, badgers, and field mice are the most common reservoirs in the region (Lines 57-59). We corrected and reference [12] cited.
Line 69-70: Authors state: Although this region had no reported cases of Q fever, it is considered as a region with a high potential of Q fever, while citing reference 11.
While reading reference 11 I could not find any information related to “high potential of Q fever’’ in Jeju Island. I ask for authors to give supportive background for their statement.
Response: Thank you for your comment. We corrected Q fever to tick-borne disease, and we cited a new reference [10] on a Q fever study in our region (Line 54).
The aim:
The aim is not in alignment with the results described in paper. Aim is oriented only toward serosurvey, while tick sampling and analysis was conducted also.
Response: Thank you for your comments. We corrected the aim including the tick analysis (Lines 62-64).
Materials and Methods
In the first step, ten rural villages were selected depending on the type of agriculture practiced and the distance from urban areas.
Are there any urban villages in South Korea? Please be more specific about criteria that you referred as “agriculture practiced” and “distance from urban areas”.
Response: Thank you for your comment City is called “-si in Korean language” in the administrative division system in South Korea.
In 2020, the resident registration population of Jeju-do was about 696,000, the largest among the islands in South Korea. Jeju-do is the only self-governing province in South Korea, meaning that the province is run by local inhabitants instead of politicians from the mainland. Jeju-do is divided into 2 cities (Jeju-si and Seogwipo-si; population more than 50,000), and 12 rural areas (7 eups and 5 myeons. The population is < 50,000 in eups and > 20,000 in myeons). Two of myeon were excluded from the analysis as sub-island area of Jeju-do. We added a description about the selection criteria and the administrative division system (Lines 73-75).
Authors stated that from 352 farmers who accepted to participate in the study 270 samples was acquired. What was the reason for other subjects to be excluded from the study? How much blood was collected and in what kind of vials?
Response: Thank you for your comment. A sample of 10 mL of blood was collected from each of the 352 participants in three plain tubes and one EDTA tube. Two hundred thirty participants’ samples had a sufficient volume of blood for multiple tests and the other were excluded because of an insufficient plasma volume. We have added these details in Lines 78-80.
Data regarding patients’ sociodemographic status, medical history, occupation, type of farming, history of tick bites, history of Q fever, history of contact with Q fever, and other relevant information were collected using standardized questionnaires.
I ask for authors to upload used questionnaire as supplementary file.
Response: Thank you for your comment. We would like to upload a questionnaire file as supplementary file, but the questionnaire is in Korean is 33 pages long. It's too long to provide as a supplementary file. If you would like a copy, we can send it to your personal mail or to the editor. What do you recommend?
The collected blood samples were preserved at -70°C after centrifugation, until further analyses.
I propose that authors referrer to centrifuged blood sample as serum or plasma, depending on vials used. How much time has been passed from blood collection to sera/plasma separation and deep freezing?
Response: Thank you for your opinion. The collected blood samples were immediately centrifuged at the interview site. After centrifugation, plasma were stored in general refrigerator until the end of the day. After returning to the hospital, the collected blood samples were frozen at -70°C in a deep freezer in our institute on the same day. We corrected this (Lines 84-86).
Line 103-104
The sensitivity and specificity of this laboratory test (Phase I and II) were 100% and 100%, respectively [13, 14].
The authors are giving misleading statements. In the assay datasheet manufacturer is referring to Relative Sensitivity and Relative Specificity. In addition, both for IgM and IgG in the manufacturer datasheet it is stated: seroconversion alone may not be diagnostic of current or recent infection.
Response: Thank you for your comments. We were not able to do sensitivity and specificity assays in this study, so, we described the results given in the manufacture reference. We corrected this to “relative” (Lines 91-92). We did not intend to study current or recent infection for seroconversion. We intended to conduct a seroprevalence survey to determine the need for diagnostic testing for Q fever among people living in the area.
Coxiella burnetii identification in ticks
- On what locations and in which part of the year the ticks were collected?
Response: Thank you for your question. From June 2016 to January 2019, well-equipped trained researchers collected ticks from the natural environment of Jeju Island. The tick sampling sites included 5 rural areas: Aewol-eup (AW); Seon Hul-ri (SH); Jeo Ji-ri (JJ); and Ha Do-ri (HD) and Bo Mok-ri (BM). Ticks were manually collected twice a month, during the first and third weeks. We changed the wording to “per month (January to December)” (Lines 96-97).
- Using which “manual” method?
Response: Yes, we additionally described “by dragging a white cloth in woodlands for 2 hours in each area” (Line 97).
- What key was used for morphological determination?
Response: Thank you for your question We morphologically identified tick species and developmental stages by using an Olympus SD-ILK-200–2 stereomicroscope (ref. Goff ML, Loomis RB, Welbourn WC, Wrenn WJ. A glossary of chigger terminology (Acari: Trombiculidae); J Med Entomol. 1982;19:221–38. https://doi.org/10.1093/jmedent/19.3.221). We inserted this as new Reference 18 (Line 99).
- What nucleic acid extraction method was used?
Response: Total Haemaphysalis longicornis nucleic acid was extracted using a QIAamp RNA Mini kit (QIAGEN Inc., https://www.qiagen.com) according to the manufacturer’s instructions (Lines 99-100).
- If 3193 ticks were collected, what was the rationality for further examination of 351 ticks?
Response: Originally, we wanted to determine the occurrence of coinfection by bacteria such as Orientia tsutsugamushi, which are bacteria transmitted to humans by chigger mite bites and severe fever with thrombocytopenia syndrome virus (SFTSV) is a tick-borne hemorrhagic fever virus, because coinfection with Orientia tsutsugamushi has already been reported (ref. Wi YM, Woo HI, Park D, Lee KH, Kang CI, Chung DR, Peck KR, Song JH, 2016. Severe fever with thrombocytopenia syndrome in patients suspected of having scrub typhus. Emerg Infect Dis 22: 1992–1995).
Therefore, after we collected 3,193 ticks, first detected SFTSV in 11.1% (354/3,193) of 3193 ticks, and we found 354 ticks were positive of SFTSV (ref. Severe Fever with Thrombocytopenia Syndrome Virus in Ticks and Incidence of Severe Fever with Thrombocytopenia Syndrome in Korea Emerging Infectious Diseases 2020, 26 (9): 2294-2296). We tried to detect bacterial 16S RNA using 16S PCR-sequencing among the 354 ticks, which are positive SFTSV and we found the Coxiella burnetii gene sequence in the ticks (Lines 100-103).
In total, 3,193 ticks were collected, 351 of whom were identified to have severe fever with thrombocytopenia syndrome viruses (SFTSV).
This sentence does not belong to MM section. In addition, I don’t know what is the relevance of SFTSV for this study?
Response: We first analyzed the ticks for a SFTSV study (Ref. Severe Fever with Thrombocytopenia Syndrome Virus in Ticks and Incidence of Severe Fever with Thrombocytopenia Syndrome in Korea Emerging Infectious Diseases 2020, 26 (9): 2294-2296), then, in the SFTSV-infected ticks, we performed RNA gene sequencing for C. burnetii. We inserted reference [14] (Lines 100-102).
Results
Figure 1 is named Geographic distribution of seroactivity participants with Coxiella burnetii in agricultural population and Coxiella burnetii-positive ticks in Jeju Island.
- Seroreactivity instead seroactivity
Response: Thank you for comment. We corrected the figure caption (Line 223).
- Authors marked with a star “ burnetti collection region in ticks’’. I can’t understand what authors wanted to say with this. Please be more specific.
Response: Our aim was to investigate the Coxiella burnetii in ticks from the five rural regions. So, we marked the regions in which ticks were collected with an asterisk. This is described in the Methods section 2.3. We additionally inserted Figure 1. We have added the latitude and longitude of the regions in which ticks were collected in the Figure legends. C. burnetii was identified in ticks in all the regions in which ticks were collected. This is described in Table 3.
As I can notice in the region map, all countries bordering South Korea except Democratic People's Republic of Korea are labeled. Please label all bordering counties.
Response: Thank you for comment. We have added a label for the People’s Democratic Republic of Korea.
Comparison of clinical characteristics between the C. burnetii-positive and -negative groups
Line 161, 162 – change term patient to participant
Response: Thank you for comment. We changed “patient” to “participant” (Line 124).
Older farmers showed a trend of a higher risk of C. burnetii infection than 163 younger farmers (p < 0.07)
If authors are referring to risks, I have to ask for them firstly to conduct a risk analysis.
Response: Thank you for comment. We deleted the word risk and reworded the sentence (Lines 130-131).
I would ask for authors to explain why are they are assuming that older farmers have a higher risk of C. burnetii infection that younger ones when there is no statistically significant difference between these two age groups?
In previous studies, older participants have had a higher prevalence of C. burnetii infection than more younger participants. There was no statically significant difference in our study. We deleted the word “risk.”
The mean Charlson comorbidity index (CCI) score did not show any significant difference… Utility of this index was not mentioned in MM section.
Response: Thank you for comment. We added a description of the CCI to the Methods and Materials section (Line 82).
Coxiella burnetii in ticks in Jeju Island
Please provide supplementary table with data related to location, time, species of tick, developmental stage of tick and molecular findings for all collected/analyzed ticks.
Response: Thank you for suggestion. We added a supplementary table. Haemaphysalis longicornis (98.5%) was the most common tick, followed by Haemaphysalis flava (1.5%). This is reported in the Results section (Line 142).
Discussion
- Physicians may find it difficult to diagnose Q fever in patients with acute febrile illness because there are no easy commercial diagnostic tools and there is a lack of seroepidemiological data in South Korea
I would ask for authors to explain this statement. As for my current knowledge - backbone of effective diagnostic approach are not easy commercial diagnostic tools, neither seroepidemiological data. Determination of endemic areas, surveillance and implementation of specific management protocols are more effective in most of the counties.
Response: Thank you for comments. We agree with your opinion. In South Korea, diagnosis of Coxiella burnetii can only be confirmed by performing the by IFA test (Phase I and II) in the Korea Disease Control and Prevention Agency (KDCPA) laboratory. Physicians send a sample to the KDCPA with the patient’s declaration form. It a few weeks to receive the result. Because of the centralized testing, surveillance of patients with Q fever is inadequate. We suggest that a commercial diagnostic tool that can be easily applied in a general hospital be made available, as this would be more efficient. We add the line 173-175.
-Authors are building discussion according to opinion that all seroreactive patients had C. burnetti infection.
As stated by manufacturer, if patient did not have specific clinical manifestations, C. burnetti infection that cannot be assumed via mentioned tests. I strongly suggest for authors to critically review their results and rewrite discussion with emphasis of explaining the current situation concerning possible exposure to C. burnetti, especially in this case where authors report to have so high seroprevalence compared to other regions/countries, thus without many actual reported Q-fever cases.
Response: Thank you for comments. We already described in the discussion section that 40,000 cases reported a low Q fever seroprevalence (2.4%) in the Netherlands, 24% seroprevalence in the acute febrile cohort in France [reference 4]. In our study, there was a significant difference in thevariable according to the orchard type, so, it was difficult to compare our results with other studies. In our previous SFTS related serologic study, the seropositive was also high in fruit farmers [Yoo et al. Seroprevalence of Severe Fever with Thrombocytopenia Syndrome in the Agricultural Population of Jeju Island, Korea, 2015-2017 Infect Chemother. 2019 Dec;51(4):337-344. doi: 10.3947/ic.2019.51.4.337.].
Author stated that:
In this study CPG was presumed to have infection due to a tick bite or droplet transmission from an infected animal because the agricultural population did not wear an N95 mask or any other personal protective equipment during the pre-coronavirus disease pandemic era, and infected domestic animals shed a large amount of C. burnetii.
Thus, in results they reported that more that 60% of seropositive participants used facial mask. Please explain this discrepancy.
Response: Thank you for your comment. We suggest that Coxiella burnetii infection is possible by other transmission modes such as tick bite, and farmers do not wear a mask all the time, and may become infected with as respiratory infection after agricultural work in their home environment. In addition, we suggest wearing with an N95 mask or any other personal protective equipment, facial mask may have a less protective effect against Q fever. Other studies have also mentioned this. We additionally described that “an N95 mask or any other personal protective equipment” in the discussion section (Lines 184-189).
According to current manuscript, this study has many limitations, while the biggest one is non-critical interpretation of serological assays.
Response: Thank you for your comment. We are aware that this study has many limitations. We examined only 1:16 and 1: 256 titer using IFA assay because serial dilutions were not analyzed. This limitation was already described in the limitation section. The study was conducted in healthy participants, so it was difficult to collect a large amount of blood. We ask you to consider this limitation.
Reviewer 3 Report
Authors of "Seroprevalence study of Coxiella burnetii in a healthy agricultural population of a rare region of Q fever in South Korea" describes prevalence in one island in Korea. The results of this study have a certain importance, describing the risk of becoming infected with Coxiella of a significant part of the rural population.
Comments:
Introduction:
Lines 52-55: "the other routes of infection transmission to humans are the consumption of contaminated milk and dairy products, skin or mucosal contact, tick bites, blood transfusion, and sexual transmission [6, 7]." I disagree with part of this assess and the references. 6 only studies wind over q fever transmission and 7 said "...is also a risk, probably through the inhalation of tick faeces, since ticks concentrate the organism in their faeces. Humans rarely, if ever, acquire disease through tick bites." I am not sure about tick bites as a risk factor fot Q fever. I recommend this manuscript 10.1016/j.pt.2015.06.014 or find a demostration about tick bites and Coxiella transmission.
Lines 68-70: Something is wrong in these two sentences "In the past 20 years, only two patients with Q fever have been reported in Jeju Island [10]. Although this region had no reported cases of Q fever, it is considered as a region with a high potential of Q fever [11]". First sentence contradicts the second. Explain it in a clear way. This means that line 29 in abstract "Although our region has no reported cases..." No one or two cases?
Somewhere between lines 61 and 64, authors should include a reason why they are sampling the ticks. To monitor the presence of Coxiella? As a demonstrated vector? because the latter I am not sure has been demonstrated for the human being (reference please)
M&M
Lines 112-113: Authors must change this sentence "In total, 3,193 ticks were collected, 351 of whom were identified to have severe fever with thrombocytopenia syndrome viruses (SFTSV)." for something like In total, 3,193 ticks were collected, 351 of whom were identified to have thrombocytopenia syndrome viruses (SFTSV), that causes haemorrhagic fever in East Asia or similar (the original said to us that tick were suffering a very severe fever).
In lines 113-115 authors said: "Clone XCP-1 16S ribosomal RNA gene sequencing was used for identifying Coxiella species in the 351 ticks that had SFTSV (Table 2)." Why only in these particular 351 ticks and not in the other? (3193 as a total) I suppose that you think that these ticks have biten people; is it so? If it is, please explain it. If not, explain to me why.
Results:
Table 1: Some of the factor categories are almost empty. Probably you can obtain better results fitting some of the categories. On the othe hand, I think that your sampling size is a bit short. I know it is difficult to convince people to participate in these studies, but you started with a good number (500) but the last was a bit scarce. Difficult to correct now; It is what it is.
Discussion:
Lines 195-196: "Only one patient in Jeju Island has been reported to have Q fever as reported in the Korean notifiable national system during the study period [12]" Again, readers may have a mess in their brain. Initially, we have two people, then none of them and know one. Please unified the number of people with q fever
Lines 216-219: "Patients with Q fever would have developed the disease without animal contact because the routes of transmission to humans would mainly be through tick bites and aerosol inhalation, influenced by geographical characteristics such as subtropical climate, specific natural environment, and heavy wind in Jeju Island [11]". Since I have never been there and I do not grow orchids, I cannot know, but what compost do they use for the plants, if they use one? Is it animal manure? Could not also the fertilizer they use a source of infection?
Author Response
Author response
Manuscript number: Manuscript ID: pathogens-1337633
Manuscript Title: Seroprevalence of Coxiella burnetii antibodies in a healthy agricultural population and prevalence of Coxiella burnetii infection in ticks of non-endemic regionregion for Q fever South Korea
Dear Main Editor and Reviewers:
Thank you for reviewing our manuscript and for your comments.
Response to your comments
Reviewer 3
Comments and Suggestions for Authors
Authors of "Seroprevalence study of Coxiella burnetii in a healthy agricultural population of a rare region of Q fever in South Korea" describes prevalence in one island in Korea. The results of this study have a certain importance, describing the risk of becoming infected with Coxiella of a significant part of the rural population.
Comments:
Introduction:
Lines 52-55: "the other routes of infection transmission to humans are the consumption of contaminated milk and dairy products, skin or mucosal contact, tick bites, blood transfusion, and sexual transmission [6, 7]." I disagree with part of this assess and the references. 6 only studies wind over Q fever transmission and 7 said "...is also a risk, probably through the inhalation of tick faeces, since ticks concentrate the organism in their faeces. Humans rarely, if ever, acquire disease through tick bites." I am not sure about tick bites as a risk factor for Q fever. I recommend this manuscript 10.1016/j.pt.2015.06.014 or find a demonstration about tick bites and Coxiella transmission.
Response: Thank you for your comment. We additionally inserted reference 8 [10.1016/j.pt.2015.06.014], and deleted tick bite, based on your comments.
Lines 68-70: Something is wrong in these two sentences "In the past 20 years, only two patients with Q fever have been reported in Jeju Island [10]. Although this region had no reported cases of Q fever, it is considered as a region with a high potential of Q fever [11]". First sentence contradicts the second. Explain it in a clear way. This means that line 29 in abstract "Although our region has no reported cases..." No one or two cases?
Response: Thank you for your comment. There have been 2 cases of Q fever reported: one in 2011 and one in 2018. We corrected to the wording based on your comments (Lines 149-150).
Somewhere between lines 61 and 64, authors should include a reason why they are sampling the ticks. To monitor the presence of Coxiella? As a demonstrated vector? because the latter I am not sure has been demonstrated for the human being (reference please).
Response: Thank you for your comment. Particularly, in our region severe fever with thrombocytopenia syndrome is the most common infectious disease in South Korea. The main vector as Haemaphysalis longicornis (our published article reference 11 and additionally supplementary table). In addition, the numbers of horses and pigs are very high. In our region, 52.4% of Coxiella positivity were positive in horses (reference 9), The seroprevalence in cattle on Jeju Island (21.3%, 95% CI: 14.5–28.0) was significantly higher than in cattle in any of the other three geographical regions (reference 10). We examined ticks to assess the potential for tick-borne transmission because of the many opportunities for exposure to ticks (Lines 53-55).
Material and methods
Lines 112-113: Authors must change this sentence: "In total, 3,193 ticks were collected, 351 of whom were identified to have severe fever with thrombocytopenia syndrome viruses (SFTSV)." for something like: “In total, 3,193 ticks were collected, 351 of whom were identified to have thrombocytopenia syndrome viruses (SFTSV), that causes haemorrhagic fever in East Asia” or similar (the original said to us that tick were suffering a very severe fever).
Response: Thank you for your comment. We corrected the sentence to: “In total, 3,193 ticks were collected, of which 354 were found to harbor severe fever with thrombocytopenia syndrome virus (SFTSV), which causes severe fever with thrombocytopenia syndrome in humans in East Asia.” (Lines 100-102)
In lines 113-115 authors said: "Clone XCP-1 16S ribosomal RNA gene sequencing was used for identifying Coxiella species in the 351 ticks that had SFTSV (Table 2)." Why only in these particular 351 ticks and not in the other? (3193 as a total) I suppose that you think that these ticks have bitten people; is it so? If it is, please explain it. If not, explain to me why.
Response: Thank you for your question. Originally, we wanted to know whether the ticks were infected with Orientia tsutsugamushi, which are bacteria transmitted to humans by chigger mite bites, and severe fever with thrombocytopenia syndrome virus (SFTSV), which is a tick-borne hemorrhagic fever virus, because coinfection with Orientia tsutsugamushi has previously been reported (ref. Wi YM, Woo HI, Park D, Lee KH, Kang CI, Chung DR, Peck KR, Song JH, 2016. Severe fever with thrombocytopenia syndrome in patients suspected of having scrub typhus. Emerg Infect Dis 22: 1992–1995).
Therefore, after we collected 3,193 ticks, we first detected SFTSV in 11.1% (354/3,193) of 3193 ticks (ref. Severe Fever with Thrombocytopenia Syndrome Virus in Ticks and Incidence of Severe Fever with Thrombocytopenia Syndrome in Korea Emerging Infectious Diseases 2020, 26 (9): 2294-2296). We then tried to detect bacterial 16S RNA using 16S PCR-sequencing among 354 ticks, which were positive for SFTSV, and we found the Coxiella burnetii gene sequence in these ticks.
Results:
Table 1: Some of the factor categories are almost empty. Probably you can obtain better results fitting some of the categories. On the other hand, I think that your sampling size is a bit short. I know it is difficult to convince people to participate in these studies, but you started with a good number (500) but the last was a bit scarce. Difficult to correct now; It is what it is.
Response: Thank you for your comment. We sincerely appreciate your understanding the limitation of this study. We do not understand your comment: “Probably you can obtain better results fitting some of the categories.”
Discussion:
Lines 195-196: "Only one patient in Jeju Island has been reported to have Q fever as reported in the Korean notifiable national system during the study period [12]" Again, readers may have a mess in their brain. Initially, we have two people, then none of them and now one. Please unified the number of people with Q fever
Response: Thank you for your comment. We corrected this sentence to “Only two cases of Q fever on Jeju Island were reported in the Korean national notifiable infectious disease system during the period 2001-2020” (Lines 149-150). The figure is given as two consistently throughout the revised manuscript.
Lines 216-219: "Patients with Q fever would have developed the disease without animal contact because the routes of transmission to humans would mainly be through tick bites and aerosol inhalation, influenced by geographical characteristics such as subtropical climate, specific natural environment, and heavy wind in Jeju Island [11]". Since I have never been there and I do not grow orchids, I cannot know, but what compost do they use for the plants, if they use one? Is it animal manure? Could not also the fertilizer they use a source of infection?
Response: Thank you for your comment. We did not consider the impact of prefectural compost. Participants were not asked about the type of compost that they used. In addition, if the farmer uses animal manure, we think the effect of compost will be sufficient. It is difficult to describe in the main text as we were unable to find any studies related to compost in the literature. However, we additionally described the lack of information on compost in the limitations section (Lines 211-212).
Round 2
Reviewer 2 Report
Authors corrected some parts of the manuscript, thus many things are still problematic.
Title: Seroprevalence of Coxiella burnetii antibodies in a healthy agricultural population and prevalence of Coxiella burnetii infection in ticks of a non-endemic region for Q fever in South Korea
There are no Coxiella burnetii antibodies in humans, only anti-Coxiella burnetii antibodies
- C. burnetii is transmitted to humans mainly via the inhalation of contaminated aerosols from infected animals. Other routes of transmission to humans include the consumption of contaminated milk and dairy products, skin or mucosal contact, ticks, blood transfusion, and sexual transmission [4, 6, 7, 8].
In reference 8 it is clearly stated that vector capacity of ticks to transmit C. burnetii is unclear. Either find reference that support your claim either rephrase the sentence according to this reference.
Among the tick 61 species, C. burnetii has been identified in Haemaphysalis longicornis, which is the dominant tick species in South Korea.
Please mention data concerning vector capacity of Haemaphysalis longicornis for C. burnetii.
Alt-72 hough reported cases of Q fever are rare in this region [13], the region is considered to 73 have a potential for Q fever to occur [9, 10], because of the very high density of H. longicornis [14] and a high prevalence of Coxiella in ticks and horses in the region [10].
Is it confirmed that H. longicornis is competent vector for C. burnetii? If not, rephrase the sentence. And rephrase every part of discussion where it is suggested that tick is the source of C. burnetii exposure for your study population and state that it is only a hypothetic source of exposure.
The present study aimed to investigate the seroprevalence of C. burnetii antibodies in a healthy agricultural population…
I assume that you wanted to examine anti- C. burnetii antibodies? Please correct this in every part of manuscript.
This study was conducted from January 2015 to December 2019 in a healthy agricultural population living in the rural areas of Jeju Island.
We see later that many participants have comorbidities. Why then you refer them as healthy?
The remaining 122 samples were excluded because of an insufficient plasma volume in the SST samples.
- You can’t separate plasma in serum tubes. It is impossible. Please correct this in every part of manuscript.
- If you have collected 3 vials for serum separation, it is not possible for you to not have enough material for detection of burnetii seroreactivity.
Testing of ticks for thrombocytopenia syndrome virus is not under the aim of this study. Either change aim or remove data related to examination of thrombocytopenia syndrome virus.
Please provide supplementary table as whole dataset concerning tick collection according to Estrada-Peña A, Cevidanes A, Sprong H, Millán J. Pitfalls in Tick and Tick-Borne Pathogens Research, Some Recommendations and a Call for Data Sharing. Pathogens. 2021;10(6):712. Published 2021 Jun 7. doi:10.3390/pathogens10060712
On Jeju Island, transthoracic echocardiography should be performed in patients with confirmed Q fever and in patients with unexplained acute fever, because Q fever can lead to severe complications.
What is the background for this recommendation? Any references?
If authors are suggesting that ‘’Patients with acute febrile illness who reside in these areas should have a diagnostic assessment for Q fever’’. I would ask for authors to mention in discussion what is the most common cause for acute febrile illness in Jeju island. Is there any differential diagnostic tool that can be used before diagnostic assessment for Q fever is initiated?
This study presents novel evidence that the incidence of Q fever is relatively high in high-risk human populations in South Korea.
I would argue with this statement. Authors are presenting possible frequent exposure to C. burnetii according to detected seroreactivity. Clinical manifestation of Q fever is not demonstrated to be ‘’relatively high’’.
I would suggest for authors to make corrections according to this comment and submit manuscript for another revision.
Author Response
Author response
Manuscript number: Manuscript ID: pathogens-1337633 rev2
Manuscript Title: Seroreactivity to Coxiella burnetii in an agricultural population and prevalence of Coxiella burnetii infection in ticks of a non-endemic region for Q fever in South Korea
Dear Editor and Reviewer 2:
Thank you for your sincere comments.
Reviewer 2
Comments and Suggestions for Authors
Reviewer 2: Authors corrected some parts of the manuscript, thus many things are still problematic.
Title: Seroprevalence of Coxiella burnetii antibodies in a healthy agricultural population and prevalence of Coxiella burnetii infection in ticks of a non-endemic region for Q fever in South Korea
There are no Coxiella burnetii antibodies in humans, only anti-Coxiella burnetii antibodies.
Response: We understand your suggestion, but other serological studies have not used the term “anti-Coxiella burnetii antibodies.” We have changed the title to “Seroreactivity to Coxiella burnetii” because your comment is technically correct. (Page 1, Line 2)
- C. burnetii is transmitted to humans mainly via the inhalation of contaminated aerosols from infected animals. Other routes of transmission to humans include the consumption of contaminated milk and dairy products, skin or mucosal contact, ticks, blood transfusion, and sexual transmission [4, 6, 7, 8].
Reviewer 2: In reference 8 it is clearly stated that vector capacity of ticks to transmit C. burnetii is unclear. Either find reference that support your claim or rephrase the sentence according to this reference.
Response: We have revised the statement about the transmission route. In addition, we added a new reference about a case of SFTS and Q fever coinfection in an 8-year-old girl following a tick bite (Kim JH, Choi YJ, Lee KS, et al. Severe fever with thrombocytopenia syndrome with Q fever coinfection in an 8-year-old. Pediatr Infect Dis J 2021;40:e31-e34.) (Page 2, Lines 54-58).
Among the tick 61 species, C. burnetii has been identified in Haemaphysalis longicornis, which is the dominant tick species in South Korea.
Please mention data concerning vector capacity of Haemaphysalis longicornis for C. burnetii.
Response: We have provided data concerning the vector capacity of H. longicornis, as you suggested, and added a new reference (Lee JH, Park HS, Jang WJ, et al. Identification of the Coxiella sp. detected from Haemaphysalis longicornis ticks in Korea. Microbiol Immunol 2004; 48:125-130.) (Page 2, Lines 66-69).
Although reported cases of Q fever are rare in this region [13], the region is considered to have a potential for Q fever to occur [9, 10], because of the very high density of H. longicornis [14] and a high prevalence of Coxiella in ticks and horses in the region [10].
Is it confirmed that H. longicornis is competent vector for C. burnetii? If not, rephrase the sentence. And rephrase every part of discussion where it is suggested that tick is the source of C. burnetii exposure for your study population and state that it is only a hypothetical source of exposure.
Response: Thank you for your question. It has not been confirmed that H. longicornis is a competent vector for C. burnetii. We have added a sentence clarifying this. (Page 2, Lines 80-81)
The present study aimed to investigate the seroprevalence of C. burnetii antibodies in a healthy agricultural population…
I assume that you wanted to examine anti-C. burnetii antibodies? Please correct this in every part of manuscript.
Response: Thank you for your comment. We have changed “seroprevalence of C. burnetii antibodies” to “seroprevalence of antibodies to C. burnetii.” (Line 82)
This study was conducted from January 2015 to December 2019 in a healthy agricultural population living in the rural areas of Jeju Island.
We see later that many participants have comorbidities. Why then you refer them as healthy?
Response: Thank you for your comment. It is expected that they will have some comorbidities because of aging. However, they did not have an acute disease, and they had common comorbidities such as hypertension and diabetes. We deleted the word “healthy.”
The remaining 122 samples were excluded because of an insufficient plasma volume in the SST samples.
- You can’t separate plasma in serum tubes. It is impossible. Please correct this in every part of manuscript.
Response: We corrected “plasma” to “serum.” (Line 106)
- If you have collected 3 vials for serum separation, it is not possible for you to not have enough material for detection of burnetii
Response: Although the study protocol specified that 3 SST tubes and 1 EDTA tube of blood should be collected, some participants provided less than the specified volume of blood. It is difficult to persuade healthy individuals to donate multiple tubes of blood for research that has no direct benefit.
Testing of ticks for thrombocytopenia syndrome virus is not under the aim of this study. Either change aim or remove data related to examination of thrombocytopenia syndrome virus.
Response: Thank you for your comment. We have removed the data on testing for severe fever with thrombocytopenia syndrome virus because it was not an aim of this study.
Please provide supplementary table as whole dataset concerning tick collection according to Estrada-Peña A, Cevidanes A, Sprong H, Millán J. Pitfalls in Tick and Tick-Borne Pathogens Research, Some Recommendations and a Call for Data Sharing. Pathogens. 2021;10(6):712. Published 2021 Jun 7. doi:10.3390/pathogens10060712
Response: Thank you for your comment and for drawing this reference to our attention. We have added a supplementary table summarizing the results of testing of ticks for C. burnetii according to their developmental stage. We tested each tick individually for Coxiella. We did not perform pooled testing.
On Jeju Island, transthoracic echocardiography should be performed in patients with confirmed Q fever and in patients with unexplained acute fever, because Q fever can lead to severe complications.
What is the background for this recommendation? Any references?
Response: The Infectious Diseases Society of America recommends transthoracic echocardiography in patients with Bartonella, Legionella, or C. burnetii infection (IDSA guideline, Infective Endocarditis in Adults: Diagnosis, Antimicrobial Therapy, and Management of Complications – A Scientific Statement for Healthcare Professionals from the American Heart Association). This recommendation is the same as the Korean guideline (The Korean Society of Infectious Diseases, Korean Society for Chemotherapy. Clinical Guideline for the Diagnosis and Treatment of Cardiovascular Infections Infect Chemother 2011;43(2):129-177 DOI: 10.3947/ic.2011.43.2.129).
If authors are suggesting that ‘’Patients with acute febrile illness who reside in these areas should have a diagnostic assessment for Q fever’’. I would ask for authors to mention in discussion what is the most common cause for acute febrile illness in Jeju island. Is there any differential diagnostic tool that can be used before diagnostic assessment for Q fever is initiated?
Response: Thank you for your comment. The differential diagnosis includes severe fever with thrombocytopenia syndrome and scrub typhus (Page 8, Lines 250-253).
We have added 3 references (Kim M, Heo ST, Oh K, et al. Prognostic factors of severe fever with thrombocytopenia syndrome in South Korea Viruses 2021;13:1-10. Yoo JR, Heo ST, Koh YS, et al. Unusual genotypic distribution of Orientia tsutsugamushi strains causing human infections on Jeju Island. Am J Trop Med Hyg 2014;90:507-510. Yoo JR, Heo ST, Kang JH, et al. Mixed infection with severe fever with thrombocytopenia syndrome virus and two genotypes of scrub typhus in a patient, South Korea, 2017. Am J Trop Med Hyg 2018;99:287-290).
This study presents novel evidence that the incidence of Q fever is relatively high in high-risk human populations in South Korea.
I would argue with this statement. Authors are presenting possible frequent exposure to C. burnetii according to detected seroreactivity. Clinical manifestation of Q fever is not demonstrated to be “relatively high.”
Response: Thank you for comments. We changed “incidence of Q fever” to “seroprevalence of antibodies to C. burnetii.” (Page 9, Line 305)
I would suggest for authors to make corrections according to this comment and submit manuscript for another revision.
Response: Thank you for reviewing our manuscript and for all comments. We have addressed all your comments. We hope that you consider the revised manuscript to be suitable for publication.
Reviewer 3 Report
no comments
Author Response
Author response
Manuscript number: Manuscript ID : pathogens-1337633 rev2
Manuscript Title: Seroprevalence study of Coxiella burnetii in a healthy agricultural population of a rare region of Q fever in South Korea
Dear Main Editor and Reviewers 3
Comments and Suggestions for Authors: no comments
: Thank you for your sincere comments and review.
Round 3
Reviewer 2 Report
/